# Vincent van Gogh's Theological Chromatology: A Critical Reader of the Bible from His Option for the Poor *Avant la Lettre*

Alex Villas Boas 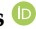

Research Centre for Theology and Religious Studies, Universidade Católica Portuguesa,
1649-023 Lisboa, Portugal; alexvboas@ucp.pt

**Abstract:** The aim of this article is to show how Vincent van Gogh developed a theological reflection that is mainly present in his paintings with religious motifs. This reflection is the fruit of his religious experience, which combines his spirituality with a social commitment to the miners in Borinage, Belgium, which can be seen as an option for the poor *avant la lettre* in the 19th century. This experience, far from strengthening his institutional relationship, rather provoked a critical attitude towards the theological discourse of the ecclesial context in which he lived and led the aspiring pastor to become a genius in painting. His theological interpretation as a critical reader of the Bible can be translated into what will be called here a theological chromatology, to be identified through the intersection of letters and paintings of Vincent van Gogh. Given the influence of the Dutch painter genius on contemporary culture, the process through which his reflection on the religious and theological issue emerges can be seen as a significant element in understanding the present in post-secular societies.

**Keywords:** Vincent van Gogh; theological chromatology; art and religion; option for the poor; post-secular societies; Émile Zola

## 1. Introduction

> "I am a friend of the poor like Jesus was"
> Vincent van Gogh

Vincent van Gogh (1853–1890) is undoubtedly one of the most influential names in Western art, and his paintings have sold for more than EUR 100 million (Burgess 2023). But what is equally (or even more) impressive is the way in which the Dutch painter, who died at the age of 37, considered a professional failure, suffering from a kind of mental illness and suspected of having taken his own life, became, within a century, a celebrated figure and a genius of painting. Moreover, he became the bearer of a profoundly Christian spirituality, as a series of books and conferences have shown, especially when the centenary of his death was celebrated in 1990.

However, this shift in the pendulum of his fame is not exactly derived from a positive valorisation of the religious issue by the author himself, as some claim, but rather indicates the need to situate him beyond a historiography that tends to completely eliminate any trace of the religious in his work and the stance of a certain religious appropriation of the painter, especially after his period of fame (Heinich 1996, p. 35s).

The religious issue in van Gogh is precisely the result of an experience of meaning, albeit critical of the religious and theological institutions of his time and context, which gives him very peculiar characteristics in terms of how he appropriates certain theological elements present in the religious motifs of his work. In this sense, within the task of an archaeology of the present time and its relationship with theology as part of the structure of thought in the West, the Dutch genius of painting results in an important excavation with regard to the critical relationship with theology, an aspect not yet explored in this unusual author for theology who concentrates in himself various elements of analysis between

spirituality and culture as a critical instance of the theology of his time, with still-significant repercussions for the present time.

Michel de Certeau, in his suggestive task of an archaeology of religion, notes how in the 19th century there was an epistemological incompatibility between the ethical primacy of the emerging human sciences, critical of the side effects of the industrial revolution, and the dogmatic primacy of the discursive practice of theology in the same period, hostage to its self-referentiality. However, it was through the social practices of the various expressions of what could be called social Catholicism and the social gospel that a kind of "Trojan horse" (De Certeau et al. 1980) was created in theology, through which dogmatic discursive practices were gradually transformed and opened up to the great dilemmas of their time and began to operate from a new ethical sensibility towards them. To this end, De Certeau seems to rely on cultural revolutions that act tactically in historical processes in order to promote a *kairos*, as an opportune moment for political and social change, but also as a task generated above all by cultural transformation. Both Foucault's strategy of the critical attitude as plural art and De Certeau's tactics operate as creative resistance, moving from frontal confrontation to installing itself as a "theology of difference" within the dominant discursive regularity and subverting it from its everyday life, silently producing a cultural revolution. For the Jesuit, the historical figures par excellence who act tactically installed in the traditional discursive regularities are those who act from the perspective of everyday life, specifically the mystic, the poet and the artist, bringing together ethics and criticism within spirituality, poetics and aesthetics, adding heterological views to culture (De Certeau 1986, pp. 171–98).

Following this path, as part of an exercise in the archaeology of the discursive practices of theology, religion can be analysed from the point of view of a complex phenomenon, and in this sense, it carries ambiguities insofar as it involves processes of the genealogy of power and the genealogy of ethics in the same phenomenon. Foucault can help to identify in Christianity, on the one hand, forms of "pastoral power" and their influence on forms of governmentality in the 19th and 20th centuries, as well as the relationship between knowledge and power in the production of theological knowledge, from the identification of its epistemological models (Villas Boas 2022, pp. 128–68), and, on the other hand, forms of "political spirituality" (Foucault 2018) that break or discontinue the theological devices for controlling behaviour.

In this context, thinking about the different epistemological ways of relating knowledge and power, madness since the 15th and 16th centuries was seen by the French philosopher as a historically constituted object, which also institutes a technology of exclusion in the way society relates to it (Foucault 2008). In this modern way of conceiving madness as mental illness, for instance, there is a differentiation from the conceptions of madness of the past, such as the Middle Ages and the Renaissance, in which this phenomenon is present in everyday life, even as an "aesthetic fact", or can even be seen as someone who tells the truth, like a "truth operator" without the will to truth or even without knowing that they tell the truth (Foucault 2019, p. 41s), a prophetic instrument (De Certeau 2015, pp. 49–73).

This modern construction of madness attributed to it an "excluded language", forbidden by its very nature as a kind of "unreason", and assumed to be inadequate to the order and logic of production inaugurated by the industrial revolution. This broad typology of madness as unreason includes libertinism, blasphemy, obstinacy in impiety or heterodoxy, practices of witchcraft or alchemy (Foucault 2019, pp. 55–56), phenomena considered to be correlates of madness, and therefore subject to internment as a form of exclusion from coexistence in order to better control the social order.

The asylum, like a madhouse, therefore, implies an orchestration of elements of truth control between a conception of scientific reason and, at the same time, theological reasons that legitimise the inclusion of religious issues (blasphemy, impiety, heresy) and moral issues (libertinism) as the basis of social exclusion techniques and the consequent practice of internment and prohibition of any expression of truth. As a consequence, the disqualification of any heterotopian discourse is instituted, in which the heretic is no longer

just someone who disagrees with the official discourse, but is seen as incapable of any relevant discourse, given his or her unreasonableness.

Vincent van Gogh himself, for instance, even without a precise diagnosis and, after his fourth psychotic break in six months at the Saint-Rémy-de-Provence asylum in France, wrote to his brother to move to the Paris suburb of Auvers-sur-Oise. There he would be close to Dr. Paul Gachet, with whom he would try a new treatment, closer to his brother, and where he could at least work instead of the "awful idleness" he was in, which for him was a real "crime". He felt that what was remained of his "reason and capacity for work" would be "absolutely in danger" if he continued to stay in the asylum (Van Gogh 1890d, cf. *Letter to Theo van Gogh*, n. 866).[1]

In this context is also where art and literature end up as languages of resistance, insofar as they lift the ban on madness (Foucault 2019, p. 59s). Art and literature are seen by the philosopher, at a certain moment, as a transgressive experience in relation to the subject of modern philosophy as a universal source of meaning and legitimiser of equally universal normative practices. In this context, modern art also becomes the language of politicised life. Foucault finds a correlation between the modern artistic experience, the experience of ancient cynicism, the medieval counter-conduct practices of Christian asceticism and the political revolutionary of the 18th and 19th centuries: the relationship between the search for a true saying and the aesthetics of existence, in which life and art merge, with art being capable of giving form that breaks with other ways of living, and life becoming the very space in which art manifests itself, artist life, even if it is marginal (Foucault [1964] 2001, p. 143).

The crossover between what can be said and what can be seen in literature and art is fundamental to understanding the turning point of a historical period, and the mutations in the knowledge of an era, which is why he evokes Borges, Cervantes, Velásquez and Flaubert, among others in his work, but with regard to art, he pays particular attention to Manet as responsible for the rupture and mutation of Western painting (Foucault 2011).

In this way, the young Vincent van Gogh lived his experience of Christian faith alongside social practices of deep solidarity with a section of the population that was very representative of the social contradictions of the Industrial Revolution. These practices were seen as inappropriate to the status quo of religious institutions, which ended up not recognising the young van Gogh's missionary vocation (Heinich 1996, p. 76). However, this experience of meaning in institutional tension begins to be dealt with through art, and runs throughout his existence, whether in his moments of lucidity or in his moments of mental disturbance. In one way or another, van Gogh's painting can be seen as a heterotopia, a practice that is developed from a non-hegemonic angle, and even discredited in the case of the works from the phase of his internment. In this sense, van Gogh is profoundly influenced by Manet (Clark 1984, pp. 23–32, 190–92), from his symbolic revolution, and according to Foucault where a new observer is born in the heterotopic space that the painter creates from the problem of lighting, that is, using not a "represented light that would illuminate the painting internally", but "the real external light" that clarifies an external light and before which, or around which, the viewer revolves by the experience of strangeness that the painting produces (Foucault 2011, pp. 31–32, 73s).

In this heterotopic dimension, van Gogh's religious motifs contribute to a heterological rethinking of some forms of theology of his time, which are still reactivated in the present, even and especially from the place of exclusion and disqualification, where they were inserted into a mechanism. It should be said that the aim of this paper is not at all to describe how the author interpreted the religious motifs in his work, but rather to identify, through the intersection of two letters and paintings, which discursive practice critically breaks with a particular model of theology, and to help think about how such a perspective and tension is relevant to understanding the present.

In this sense, too, Foucault's notion of "author" does not primarily refer to a concrete individual, but to a "position" and a "function" that a person occupies at a given moment. The "author-function" is thus that of a founder of discursivity who articulates the universe

of discourses and can simultaneously give rise to different characters, subject-positions, including different groups of individuals who may come to occupy them (Foucault 1994, pp. 803–4).

In this sense, the discursivity and visibility that emerge from van Gogh's work have theological implications that can be taken up by the critical reflection of theologians and religious scholars in order to reflect on the structural reminiscences of what is pointed out as criticism, emerged from the intersection of his religious experience and his social engagement with the miners of the Borinage, Belgium, in the 19th century. This experience can be seen as an option for the poor *avant la lettre*, bearing in mind that the use of this expression did not become common until the 20th century (Villas Boas and Sienna 2018).

However, van Gogh's experience and the way in which his reflection on it is mirrored in his work seem to indicate that this experience of intellectual responsibility for the problems of modernity, since the outbreak of the industrial revolution, is fundamental for the re-signification of the religious phenomenon and theological production in the following centuries. Above all, in what are now called post-secular societies, characterised by the re-insertion of religious communities into the public sphere, it requires the incorporation of a critical attitude, while questioning the power effects of discourses of truth (Foucault 1990, pp. 35–63), towards theology that involves overcoming the self-referentiality of the various epistemological models of theological thought (Teixeira et al. 2022). In this sense, it is possible to identify the emergence of a political spirituality in van Gogh's religious experience, which encountered a series of institutional resistances and was re-signified in an artistic expression with profound theological intuitions but aimed at an audience not limited to the religious, as the fame of his works attests. In the same way, his works have great relevance in the dialogue between spirituality and culture through the crossover of art, literature and theology.

For Foucault, spirituality is not reduced to the religious dimension, although this is a kind of historical welcoming structure for different forms of spirituality. For the philosopher, spirituality concerns the practices of self-transformation that stem from a hermeneutics of the self (Foucault 2005, p. 18) in which a new subjectivity emerges in the task of constituting a being other than oneself, in the midst of the relations of knowledge and power that constitute the discursive practices that traverse the processes of subjectivity production. This emergence takes place in an agonised process of breaking the subjectivation promoted by an apparatus established by power structures, whether political, religious or social. In other words, political spirituality is about enabling individuals to become agents of change, to move beyond conventional roles defined by authority, to forge their own identities and to consolidate a new collective consciousness. In the words of the philosopher:

> "No longer to be a subject as it has been until now, a subject in relation to a political power, but a subject of a knowledge, subject of an experience, subject also of a belief. For me, this possibility of insuring oneself the position of a subject that was fixed to him by a political power, a religious power, a dogma, a belief, a habit, or a social structure, is spirituality".
>
> (Foucault 2018, p. 21)

In van Gogh's case, the event in which the practice of a political spirituality emerges takes place at the intersection of a political agenda that emerges from the inhuman condition of the Borinage miners and the religious agenda that is restricted to pastoral work with those considered elected, which in the 19th century, in the lens of Max Weber (2001), progressively focuses on those who gather the signs of election, namely those who prosper. The work of van Gogh can be read in this sense within the notion of political spirituality as someone who constructs his subjectivity from the agonised relationship between political resistance and the faith of those who, in the context of the 19th century, were considered by Calvinist Protestant ethics and the capitalist spirit to be predestined to damnation. The paintings of van Gogh can be seen as a clear expression of a critical attitude, which is at the same time the threshold that would anticipate the archaeological soil from which the theologies of praxis would emerge.

The life and work of van Gogh can be an example of the art of not being governed in a certain way, in the face of the ecclesiastical—institutional—insistence on the government of behaviour that belongs to pastoral power as re-signified in modernity. Therefore, his life and work exemplify the notion of a critical attitude, since the criticism is not made "from outside" and is not reduced to an analysis of truth, but within the framework of pastoral government itself and the power effects of its discourse, which he himself faces from the political spirituality that his works express.

## 2. Van Gogh Effect: From Interdict to Hagiography

The sociologist specialising in contemporary art Natalie Heinich called the "van Gogh effect" the constitution of an "anthropology of admiration", which radically changes the view about highlighting his inscription in a Christian tradition. From an aesthetic point of view, Heinich presents how the artist is situated by his contemporaries in an "aesthetic guerrilla war" that disputed who would hold the representation of modernity, with the Dutch painter being at the same time praised by some and discredited by opponents in this struggle by the representation of the avant-garde (Heinich 1996, p. 9).

From the point of view of the religious resignification of the painter's biography, one would have to wait until the beginning of the twentieth century for van Gogh's religious sentiments to cease to be disqualified by the normative theologies of the nineteenth century, which were markedly moralistic. Heinich even speaks of a "hagiography" of the painter, which places him within the model of a prophetic aesthetic committed to the forgotten and exploited poor of the Industrial Revolution. The profoundly enlightening insights of a modern spirituality were seen as "bolts of lightning" amidst the turbulent storms of his biography (Heinich 1996, pp. 12, 15). It is then that van Gogh wins a laudatory litany as a "mystical soul", as someone who "dreamed the impossible", who dedicated himself to the "apostolate of beauty", as someone who had a divine "vocation to be an artist", with a great "love of nature": a "rare" painter, misunderstood by his time. Pascal Bonafoux, a French art historian, was the first to present van Gogh as an "apostle", in which his paintings would be a kind of doctrinal preaching:

> "Vincent paints as he prays and preaches. He prays and preaches in spite of the sinner that he is, just as he paints in spite of the failures of his life as a man. Holy scriptures and colours are the anguished and frenzied search for the same light, the same salvation. He is a painter as he wished to be a pastor. Vincent entered into painting without reservations, a missionary and a martyr. And his portraits are the glorious body that his painter's faith invents [...] Painting is a gift of oneself like preaching. Vincent never ceases to be a missionary and a painter, he is an apostle, i.e., a witness. Painting is God's word. And Vincent has in mind a kind of church."
>
> (Bonafoux 1990, p. 14)

This hagiographic movement about van Gogh as a kind of "Christianization of van Gogh's biography" is based mainly on exchanged letters between his brother Theo van Gogh, his sister Willemien van Gogh and the French painter and friend Émile Bernard (1868–1941). Heinich also indicates some fundamental traits of this "apostolic calling" by van Gogh that constitute this hagiography movement in which an anthropology of admiration was outlined. Topically, the sociologist points out these traits from statements in the letters exchanged by the painter with his brother and friends: (1) Need for religion: "That does not prevent me from having a terrible need of—shall I say the word?—of religion. Then I go out at night to paint the stars (Letters to Theo, 543, September 1888, III, p. 56)";[2] (2) Search for God: "To try to understand the real significance of what the great artists, the serious masters, tell us in their masterpieces, that leads to God: one man wrote or told it in a book; another, in a picture (ibid., 133, July 1880, I, p. 198)"; (3) Faith and hope: "And the moral of this is that it's my constant hope that I am not working for myself alone. I believe in the absolute necessity of a new art of colour, of design and—of the artistic life. And if we work in that faith it seems to me there is a chance that we do not hope in vain (ibid., 469,

1888, II, p. 533)"; (4) Reaching for infinity: "If what one is doing looks out upon the infinite, and if one sees that one's work has its raison d'être and continuance in the future, then one works with more serenity (ibid, 538, III, p. 39)"; (5) Dedication to work: "Today again from seven o'clock in the morning till six in the evening I worked without stirring except to take some food a step or two away. That is why the work is getting on fast (ibid., 541, 1888, III, p. 48)"; (6) Inspiration (compulsion): "I simply couldn't restrain myself or keep my hands off it or allow myself any rest (ibid. 225, August 1888, I, p. 437)"; (7) Inspiration (possession): "I have a terrible lucidity at moments, these days when nature is so beautiful, I am not conscious of myself any more, and the picture comes to me as in a dream (ibid., 543, September 1988, III, p. 58)" (Heinich 1996, p. 39).

The work of the Irish Jesuit Patrick Heelan (1926–2015) could also be situated as an echo of this movement of Christianization of the Flemish painter. The Jesuit identify in van Gogh's view a combination of his aesthetic feeling and his capacity to contemplate the world as constituting an interesting kind of non-Euclidean metaphysics. His sight of reality would allow one to think about from a systematic distortion of the mathematical structures on the representation of reality guided by a certain vision of Being (Heelan 1972, pp. 478–92). However, Heelan does not address the painter's critical relationship with religious institutions and theological debates, which could be correlated with such an exercise in non-Euclidean contemplation.

What is common among these very interesting research works is that they base their analysis mainly on the letters exchanged by van Gogh. Curiously, little or almost nothing is said about his paintings with religious motifs. And they are even considered irrelevant to the understanding of the religious vision of the painter, when comparing a few works by van Gogh with dozens of them by his friend Émile Bernard. At best, his religious work is viewed only as an expression of his piety (Mooney 2016).

The proposal here is to correlate the paintings with religious motifs and the contents of the letters as a way to better identify the critical theological hermeneutics of Vincent van Gogh as a backdrop his works.

Contrary to what van Gogh's hagiographic movement points out, this cross between the letters and the religious motifs in his painting seems to indicate a more heterological view of theology and critical theological perception of the painter in the very dynamics of the works.

### 3. Van Gogh's Religious Experience

In 1877, van Gogh was 24 years old, and he decided to study theology in order to become a Reformed Church pastor, as his father and grandfather were. The young aspiring pastor was living in Amsterdam in 1877–1878 to prepare for his examinations to become a theology student.

In a letter to his brother Theo van Gogh, in the same year, he expressed how much he wanted to follow the path of his father and grandfather, as well as following the theological references of both: "it is my prayer and deepest desire that the spirit of my Father and Grandfather may rest upon me, and that it may be given me to be a Christian and a *Christian labourer*". Said the young Vincent, paraphrasing the passage from the Book of Ruth 1:16: "Their God shall be my God, and their people my people" (Van Gogh 1877b, cf. *Letter to Theo*, n. 109).

The expression *Christian labourer* or *Christian workman* [*De christen-werkman*] refers to the Groningen School of Theology, which the father of the van Gogh belonged to, the Reverend Theodorus van Gogh, as well his brothers (Jansen et al. 2009b, cf. *The family and children of the Reverend and Mrs van Gogh*). In the 19th century, Groningen theologians themselves made an attempt to overcome the theological rationalism that arose from the doctrinal dispute between Catholic and Protestant scholastic theologies, emphasizing the experience of faith, an understanding of dependence on God rather than on dogmas, as well as commitment to the poorest (De Groot 1855, p. 45). In this way, Groningen was responsible for introducing Schleiermacher's theology into the Reformed Church in the

Netherlands, as well as valuing the humanism of Erasmus of Rotterdam and the spirituality of Thomas van Kempen, elements that made up the initial theological environment of the young aspiring pastor.

Also, van Gogh seems to have been greatly influenced by the work of one of the theologians of the Dutch Reformed Church, Ottho Gerhard Heldring (1804–1876), mentioned in at least three letters he exchanged with his brother (Van Gogh 1876a; 1877d, cf. *Letter to Theo*, nn. 096, 109, 123; 1879). His work *De Christen-werkman als zendeling* [The Christian workman as missionary] (Heldring 1847) proposes an orthodox Christianity, but one that was able to understand its social vocation in the context of the Industrial Revolution.

Nevertheless, the reasons of van Gogh repeatedly received for failing his entrance exams to study theology or become a pastor seem to have considerably altered the question of the importance of Protestant orthodoxy. When he had failed in the admission at the Faculty of Theology, he went through a preparatory "missionary" course in Brussels, and then, he decided to work along with the coal miners in Borinage, on the Belgium–French border, after November 1878, first in a village called Wasmes, and after in "Little Wasmes", living closer to the coal mine, for about one year. However, if from Heldring's perspective social engagement stems from orthodoxy, in van Gogh's such engagement resulted in a growing distancing from orthodoxy, either because of personal tensions or because of the theological inadequacy of helping the lives of the people. In April 1879, Vincent van Gogh wrote a letter to his brother telling how the coal miners' life had a shocking impact on him and the need to live like them in order to win their trust:

> [. . .] All around the mine are poor miners' dwellings with a couple of dead trees, completely black from the smoke, and thorn-hedges, dung-heaps and rubbish dumps, mountains of unusable coal etc [. . .] Some of the miners work in the *maintenages*, others load the loosened coal into small wagons that are transported along rails resembling a tramway. It's mostly children who do this, both boys and girls [. . .] If anyone were to try and make a painting of the *maintenages*, that would be something new and something unheard-of or rather never-before-seen [. . .] The people here are very uneducated and ignorant, and most of them can't read, yet they're shrewd and nimble in their difficult work, courageous, of rather small build but square-shouldered, with sombre, deep-set eyes. They're skilled at many things and work amazingly hard [. . .]. With charcoal-burners one must have a charcoal-burner's nature and character, and no pretensions, pridefulness or imperiousness, otherwise one can't get on with them and could never win their trust [. . .]. In one house everyone is sick with fever, and they have little or no help, which means that there the sick are taking care of the sick. 'Here it is the sick who nurse the sick,' said the woman, just as it is the poor who befriend the poor. [. . .]. Going down in a mine is an unpleasant business, in a kind of basket or cage like a bucket in a well, but then a well 500–700 metres deep, so that down there, looking upward, the daylight appears to be about as big as a star in the sky."
>
> (Van Gogh 1879, *Letter to Theo van Gogh*, n. 151)

In that circumstance, the *Imitation of Christ* could be pointed to as a source of his spirituality (Van Gogh 1876b, cf. *Letter to Theo*, n. 97) which could be identified in van Gogh's increasing wish to act like Jesus among the poorest. Mr. Bonté, an old neighbourhood pastor, describes his memories of the young missionary candidate in a letter from 1889. In favour of the details of his missionary work, the following long [and rich] quotation about this impression of van Gogh's dedication on the pastoral work will be used:

> "He expressed himself in French correctly and was able to preach quite satisfactorily at the religious gatherings of the little Protestant group [. . .] he considered the accommodation far too luxurious; it shocked his Christian humility, he could not bear being lodged comfortably, in a way so different from that of the miners. Therefore, he left these people who had surrounded him with sympathy and went

to live in a little hovel [. . .]. The fine suits he had arrived in never reappeared; nor did he buy any new ones. It is true he had only a modest salary, but it was sufficient to permit him to dress in accordance with his social position. Why had the boy changed this way? Faced with the destitution he encountered on his visits, his pity had induced him to give away nearly all his clothes; his money had found its way into the hands of the poor, and one might say that he had kept nothing for himself. His religious sentiments were very ardent, and he wanted to obey the words of Jesus Christ to the letter. He felt obliged to imitate the early Christians, to sacrifice all he could live without, and he wanted to be even more destitute than the majority of the miners to whom he preached the Gospel [. . .]. He preferred to go to the unfortunate, the wounded, the sick, and always stayed with them a long time; he was willing to make any sacrifice to relieve their sufferings. In addition, his profound sensibility was not limited to the human race. Vincent van Gogh respected every creature's life, even of those most despised. A repulsive caterpillar did not provoke his disgust; it was a living creature, and as such, deserved protection. The family with whom he had boarded told me that every time he found a caterpillar on the ground in the garden, he carefully picked it up and took it to a tree. Apart from this trait, which perhaps will be considered insignificant or even foolish, I have retained the impression that Vincent van Gogh was actuated by a high ideal: self-forgetfulness and devotion to all other beings was the guiding principle which he accepted wholeheartedly."

(Auden 1961, p. 62)

In another letter written by the son of the family who received him in Borinage, one can find a question asked by his mother to van Gogh about his attitude: "Monsieur Vincent, why do you deprive yourself of all your clothes like this—you who are descended from such a noble family of Dutch pastors? He answered: "I am a friend of the poor like Jesus was" (Auden 1961, p. 64). His dedication to evangelizing is also remarkably described:

"Yet he was always at his studies; in a single night he read a volume of 100 pages; during the week he taught a school he had founded for the children teaching them to fear God [. . .]. Van Gogh made many sensational conversions among the Protestants of Wasmes. People still talk of the miner whom he went to see after the accident in the Marcasse mine. The man was a habitual drinker, "an unbeliever and a blasphemer", according to the people who told me the story. When Vincent entered his house to help and comfort him, he was received with a volley of abuse. He was called especially a *rosary chewer* [*mâcheux d'chapelets*], as if he had been a Roman Catholic priest. But Van Gogh's evangelical tenderness converted the man."

(Auden 1961, pp. 64–65)

Given the precarious working conditions of the coal miners, sometimes strikes broke out, and the workers would no longer listen to anyone except "l'pasteur Vincent", the only one whom they trusted. A socialist senator from the province of Borinage later wrote to Vincent's brother how "in order to prevent bloodshed, Vincent intervened and used his great moral authority" averting clashes between the miners and the police (Auden 1961, pp. 66–67):

Nonetheless, despite the great recognition he had from the mining community, the report about the Vincent's probatory period as a missionary from the Union of Protestant Churches in Belgium in 1879 was not favourable for his admission, due to an alleged lack of a 'talent for speaking', even with the good recommendations of Mr. Bonté and several conversion cases.

"The experiment of accepting the services of a young Dutchman, Mr. Vincent van Gogh, who felt himself called to be an evangelist in the Borinage, has not produced the anticipated results. If a talent for speaking, indispensable to anyone placed at the head of a congregation, had been added to the admirable qualities

he displayed in aiding the sick and wounded, to his devotion to the spirit of self-sacrifice, of which he gave many proofs by consecrating his night's rest to them, and by stripping himself of most of his clothes and linen in their behalf, Mr. Van Gogh would certainly have been an accomplished evangelist. Undoubtedly it would be unreasonable to demand extraordinary talents. But it is evident that the absence of certain qualities may render the exercise of an evangelist's principal function wholly impossible. Unfortunately, this is the case with Mr. Van Gogh. Therefore, the probationary period—some months—having expired, it has been necessary to abandon the idea of retaining him any longer".

(Auden 1961, p. 66)

Furthermore, since his preaching was praised by the locals, the alleged lack of talent for speaking and the absence of certain qualities may have to do with not fitting in with the preaching of Calvinist doctrine. As Vincent van Gogh's friends notice, "the humanity of our friend continued to grow day by day, and yet the persecutions he suffered grew, too", referring him to reproaches and insults and 'stoning' by the members of the Reformed Church Consistory, accusing van Gogh of something between the overzealousness and the scandal. What is more, little is known about how van Gogh's family reacted to being rejected from the ministry, as they came from a family of pastors. On the one hand, his father continued his ministry as a pastor and, on the other, he continued to finance the living expenses of his son, who remained in Borinage for a while, even after the painful and traumatic experience of being rejected as a missionary (Auden 1961, pp. 64–65).

In this experience, two pillars could be pointed in the way of this young aspiring missionary contemplating the world: On the one hand, a careful reading of the biblical text in different translations, to the point that his father worried about van Gogh spending all his money on translations of "Bibles and the New Testament". On the other hand, even though he was busy with all his missionary work, his literacy of the human condition was nourished through his exercise of empathically approaching the misery of people, which translates into the insistent exercise of drawing what he contemplated (Auden 1961, pp. 61, 65). These two angles were connected by the exercise to love what the thought contemplates as the way to leads to God, and this perception was still strong in van Gogh that despite the traumatic experience with the leadership of his religious congregation, he had a rich experience of being involved in the everyday life of the Borinage community. Unlike the ecclesial trend at the time, which saw knowledge of God as a sign of divine grace that was confirmed in the clarity of interpreting the biblical scriptures according to the doctrine preached by the Church, the reprobate aspiring pastor perceived knowledge of God through reading the Bible that led him to greater empathy and engagement with the lives of the people:

"I'm always inclined to believe that the best way of knowing God is to love a great deal. Love that friend, that person, that thing, whatever you like, you'll be on the right path to knowing more thoroughly, afterwards; [. . .]. But you must love with a high, serious intimate sympathy, with a will, with intelligence, and you must always seek to know more thoroughly, better, and more. That leads to God, that leads to unshakeable faith [. . .] Someone will have attended, for a time only, the free course at the great university of poverty, and will have paid attention to the things he sees with his eyes and hears with his ears, and will have thought about it; [. . .]. Try to understand the last word of what the great artists, the serious masters, say in their masterpieces; there will be God in it. Someone has written or said it in a book, someone in a painting. And quite simply read the Bible, and the Gospels, because that will give you something to think about, and a great deal to think about and everything to think about, well then, think about this great deal, think about this everything, it raises your thinking above the ordinary level, despite yourself. Since we know how to read, let's read, then!"

(Van Gogh 1880, cf. *Letter to Theo*, n. 155)

There seems to be a parabolic dynamic in the contemplation exercise of life looking insistently and empathetically at the misery of the world, with the lens of biblical scriptures translating into drawings, a spiral dynamic which in one dimension sheds light on the other. Thus, what for some seemed a mere hobby was an exercise in finding beauty where it was not evident, as can be read in the testimony of those who lived with the young van Gogh. In the words of Mr. Bonté:

> "He would squat in the mine fields and draw the women picking up pieces of coal and going away laden with heavy sacks. It was observed that he did not reproduce the pretty things to which we are wont to attribute beauty".
>
> (Auden 1961, p. 62)

### 4. From Theologian to Painter

Émile Bernard (1868–1941), a good friend of van Gogh, who shared his religious concerns with him, was the first one to indicate that in painter there was a "transfer to painting of his religious feeling" (Heinich 1996, p. 38).

Before Borinage (1879–1880), for the aspiring pastor "truly life is a fight" (Van Gogh 1877e, cf. *Letter to Theo*, n. 133), and this concept is in line with the notion that the strength to fight came from God through mysterious ways of uniting people: "How we are bound to one another through God by bonds that are in God's hand, and in those bonds lies our strength, and they are old and do not break easily" (Van Gogh 1877c, cf. *Letter to Theo*, n. 119).

After Borinage, his ministerial vocation died, but something of the theologian still remained, transformed by his own experience, which can be seen in the distance he took from the forms of theodicy that use divine reasons to justify historical and social situations, so strong in Christian times. The coexistence of van Gogh with the miners of Borinage seems to have reinforced his conviction that life is a struggle, and that the struggle for just things is an opportunity to learn about the power of God, to believe in human goodness and the possibility of transforming one's own condition. In this ability to perceive the beauty of life, there is also a clear rejection of any form of theodicy, in attributing evil to a divine cause, as would have been customary in 19th century Calvinist scholasticism:

> "[…] everything in men and in their works that is truly good, and beautiful with an inner moral, spiritual and sublime beauty, I think that that comes from God, and that everything that is bad and wicked in the works of men and in men, that's not from God, and God doesn't find it good"
>
> (Van Gogh 1880, cf. *Letter to Theo*, n. 155)

After Borinage is also the time when the theologian became a painter, who gave colour to his drawings, and initially, in some works, he seemed to link the darker colours to the suffering life and hard work. Painting is an exercise in overcoming challenges, in which the inspiration for action comes from desire, and painting is precisely the bridge that allows this to happen. Said the painter:

> "What is drawing? How does one get there? It's working one's way through an invisible iron wall that seems to stand between what one *feels* and what one *can do*. How can one get through that wall?—since hammering on it doesn't help at all. In my view, one must undermine the wall and grind through it slowly and patiently. And behold, how can one remain dedicated to such a task without allowing oneself to be lured from it or distracted, unless one reflects and organizes one's life according to principles? And it's the same with other things as it is with artistic matters. And the great isn't something accidental; it must be *willed*" [author's emphasis].
>
> (Van Gogh 1882, cf. *Letter to Theo*, n. 274)

Drawing was a way to connect the painter with the community (Figures 1 and 2):

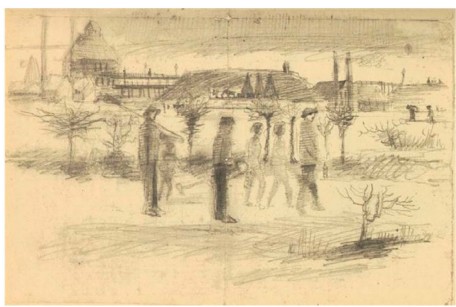

**Figure 1.** Miners in the Snow at Dawn, 1880.

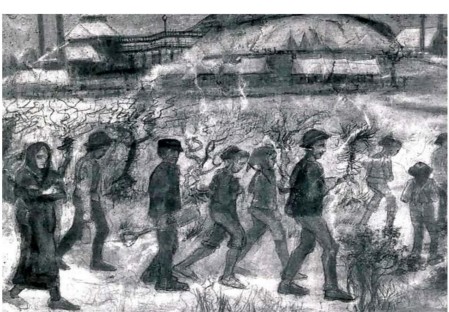

**Figure 2.** Miners in the Snow.

Etel Adnan suggests thinking of colours as a language in van Gogh (Adnan 2022) in a movement that progressively advances from darkness to light, or in the words of the painter himself: "in COLOUR seeking life, the true drawing is modelling with colour [. . .] And so I am struggling for life and progress in art" [uppercase by author] (Van Gogh 1886, cf. *Letter to Horace Mann Livens*, n. 569). The movement of painting as colourisation seemed to correspond to the exercise of consolidating the will to live and struggle for life. The painting was a kind of mediation between the feeling as a will to meaning and the action that consolidates the fight for life, a process that is lived both as an existential process and as a service to the people it connects.

He began painting in 1881. In this first phase, there were some "still life" (*nature morte*) style paintings, in a darker chromatology, but developing the technique of colourisation with the life of miners and peasants as his main motif, and some skeletons. In this period, *Still Life with Bible* (Figure 3) or simply *The Bible* was the first work with religious motif painted in 1885 in a *still life* style, on the occasion of his father's death and taking back the Bible that belonged to him.

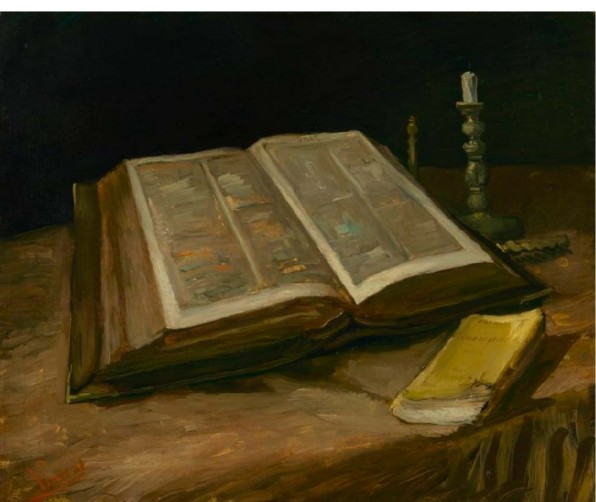

**Figure 3.** Still Life with Bible.

As well as honouring his father, the painting was also a kind of manifesto about his theological view of the Bible. On the one hand, van Gogh seems to be emphasizing a view in an instant, by means of a suppression of movements, and therefore, an anti-narrative abstract art in which the horizon is eliminated, situated in a dark space with an unlit candle, with the only textual reference of Chapter 53 (Chap LIII) of the book of Isaiah (ISAIE), called the "Song of the Suffering Servant", the Bible not being a source of illumination on its own, which for van Gogh means a source of colourisation, which in turn is a source of life.

On the other hand, the light, in a modern Manet style, comes from the left side to the right side, putting the Bible motif in the darker side, and at the same time beside a lighter book, with a growing presence of light that falls on the work *The Joy of Living* (*La Joie de vivre*) by Émile Zola (1840–1902), published in 1884. The play of colours suggests that the work establishes a geographical contiguity in a temporal continuity between the viewer and the picture. The picture seems to be painted to be an invitation to think about the present time in the instant between what *still lives* in dark colours and unable to illuminate the life of the viewer, however, not totally unable to be read, due to the light that is cast upon it, in dialogue with another space that suggests its source of light.

Zola's book's main subject is the joy of living despite the "superior forces" that threaten life, represented in particular by illness and the sea with its storms, as an expression of the power of nature that frighten the daily life of a fishing village. The main characters are the girl *Pauline*, an orphan since she was 10 years old, but who loves life despite its tragedies, and *Lazarus*, a boy deeply marked by the fear of death. There is a contrasting relationship between her optimism and his depressive pessimism, in which the joy of living is unveiled in the continuous struggle for life despite its tragedies, sufferings and unhappiness, the human force being fragile in the face of "superior forces" and a heroic insistence on living.

In another 1888 picture, van Gogh painted the *Oleanders* (Figure 4) as life-affirming flowers that bloomed "inexhaustibly", again placing the same book on the table, but in an unsteady balance, seeming to suggest Zola's condition of life: the inexhaustible will to live in precariously balanced conditions.

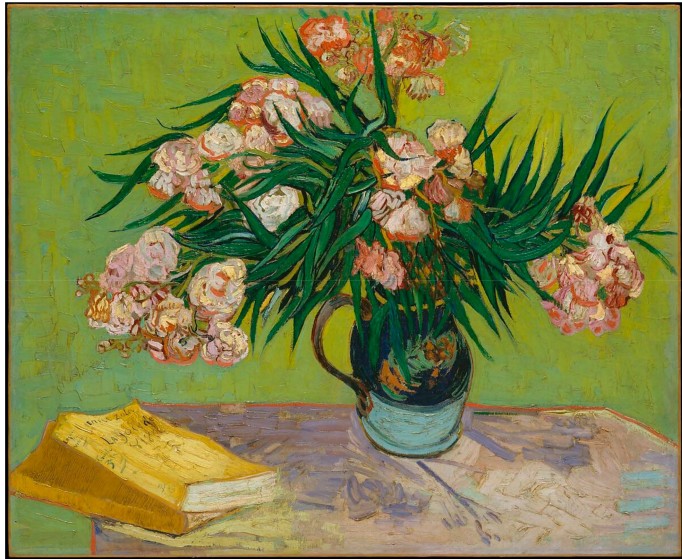

**Figure 4.** Oleanders.

In *The Joy of Living* there is also the religious reference of the Father Horteur (*l'abbé Horteur*), described by Zola in his character preparation sheets as someone who rarely spoke of God, reserving Him for his personal salvation, indifferent to other people's lives. Still, in Zola's annotations about the character one can find the main characteristics of the author's critique of religion:

"Stuck, headstrong, to the narrow formulas of Catholicism. A *religion of police and good order* [*religion de police et de bon ordre*], transmitting dogmas as instructions; life only drained him, giving him if not tolerance, at least indifference. First, I can show him still waging war against the villains who are his parishioners; then, in the end, [he] let them go: Too bad for them, they'll go to hell. And he continues to believe, makes his salvation selfish.—He no longer holds the position of a priest, except as a civil administrator" [*free translation*]

(Zola 1883, f° 245–46)

Father Hourteur with his *police religion* is the very incarnation of Foucault's pastoral power, and in this sense, *The Bible* could be thought of in this anti-narrative scenario from a crisis of capacity for meaning, being between the apparent insufficiency of the narrative of the servant's suffering and the search for the joy of living, despite the absurdity of life.

Zola is one of van Gogh's favourite authors, cited more than 90 times in his letters, being for the painter a type of literature for those who "want a truth", or more precisely, those who want to see "life as it is". Vincent van Gogh seems to see in Zola, and perhaps precisely in Pauline, a Christological trait of betting on life, as can be seen in the same letter to his sister, in which he talks about Zola and makes a comment on the Bible: "Is the Bible enough for us? Nowadays I believe Jesus himself would again say to those who just sit melancholy, *it is not here, it is risen. Why seek you*[3] *the living among the dead?* [sic]". For the author, it seems to be the modern literature that restores him from having "completely lost all inclination to laugh" for many years in his life. Literature allows him to still see the Bible as "the light of the world" and as still containing "something as great and as good and as original, and just as capable of overturning the whole old society as in the past [. . .], compare it with the old upheaval by the Christians" (Van Gogh 1887, cf. *Letter to Willemien van Gogh*, n. 574). In this perception of living as a way to fight against what he considered the evil of his time, melancholy and pessimism, in the same letter in which he praised the Bible and literature, he expresses the correlation between the task of living life intensely and the task of representing that intensity in strong, intense colours in painting, in the recommendations he made to his sister, who was thinking of becoming a painter:

"My dear little sister learn to dance or fall in love with one or more notary's clerks, officers, in short whoever's within your reach; rather, much rather commit any number of follies than study in Holland, it serves absolutely no purpose other than to make someone boring[4] [. . .] in earlier years, when I should have been in love, I immersed myself in religious and socialist affairs and considered art more sacred, more than now. Why are religion or law or art so sacred? People who do nothing other than be in love are perhaps more serious and holier than those who sacrifice their love and their heart to an idea. Be this as it may, to write a book, to perform a deed, to make a painting with life in it, one must be a living person oneself. And so, for you, unless you never want to progress, studying is very much a side issue. Enjoy yourself as much as you can and have as many distractions as you can, and be aware that what people want in art nowadays has to be very lively, with strong colour, very intense. So, intensify your own health and strength and life a little, that's the best study."

(Van Gogh 1887, cf. *Letter to Willemien van Gogh*, n. 574)

Still in the same letter, he presented his fears about the theological understanding of providence, which was similar to Voltaire's criticism in *Candide*, also mentioned by the painter in several letters, in yet another clear rejection of theodicies that would attribute divine causes to human issues:

"I feel uneasy about assuming for my own use or recommending to others for theirs the belief that powers above us intervene personally to help us or to comfort us. Providence is such a strange thing, and I tell you that I definitely don't know what to make of it."

(Van Gogh 1887, cf. *Letter to Willemien van Gogh*, n. 574)

This seems to be the same context in which, years before van Gogh painted *The Bible*, he expressed in an exchanged letter with his brother Theo van Gogh on 23 December 1881 his feelings about his father's Christmas sermons to come. One could see something about that theological academic perspective of the Bible, and simultaneously about life, evoking the need for a more emphatic perspective to reality, as the poets and the painters see the world:

> "I [also] find Pa and Ma's sermons and ideas about God, people, morality, virtue, almost complete nonsense. I also read the Bible sometimes, just as I sometimes read Michelet or Balzac or Eliot, but I see completely different things in the Bible than Pa sees, and I can't agree at all with Pa makes of it in his petty, academic way. [...] And they certainly understand the Bible just as little. Take Mauve (his painting teacher), for instance, when he reads something deep he doesn't immediately say, that man means this or that. Because poetry is so deep and intangible that one can't simply define it all systematically [...]. I find that sensibility to be worth so much more than definition and criticism. And when I read [some Literary authors] I do so because they look at things more broadly and milder and with more love than I do."

<div align="right">(Van Gogh 1881, cf. <i>Letter to Theo</i>, n. 193)</div>

In short, reading the Bible should lead its reader to an ever greater love of life and people, and for this reason, contemplating life with a greater love for it demands a poetic view, as van Gogh told his sister years later: "For my part, I'm always glad that I've read the Bible better than many people nowadays" (Van Gogh 1887, cf. *Letter to Willemien van Gogh*, n. 574).

### 5. In *Colour*, Seeking God in Life

A little earlier, before he decided to enter the Saint-Paul-de-Mausole Asylum in Saint-Rémy, South of France, while contemplating the olive trees, he had a terrible feeling of anguish that echoed from the correlation to wake up with the suffering of Christ in the olive garden and the atrocity of Christ carrying the cross. He even criticizes the painting by his friend Émile Bernard, *Christ in the Garden of Olives* (Figure 5), as a setback for reproducing a "medieval tapestry" of horrors incompatible with "modern reality" and the need for a consoling spirituality by art (Van Gogh 1889h, cf. *Letter to Émile Bernard*, n. 822).

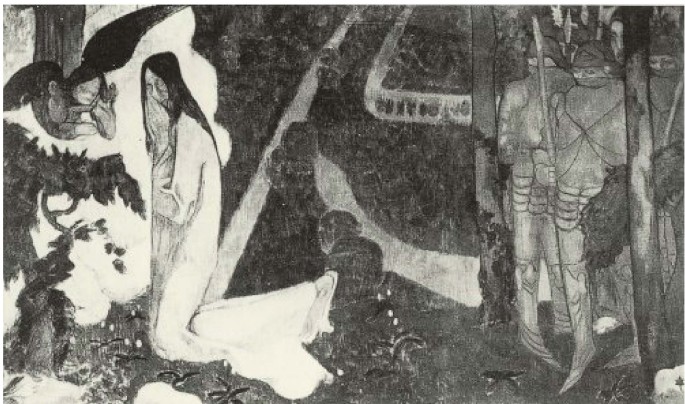

**Figure 5.** Emile Bernard, Christ in the Garden of Olives, 1889.

In the context of his mental health at the time, with four crises in which he had no idea what he "said, wanted, did", and struggling with melancholy and the desire to take his own life, the religious question itself was a challenge to discern by himself. The painter evokes the presence of priests and nuns in the asylum, which reinforces the religious imagery he was fighting against, while living with some nuns "very willingly cultivates these unhealthy religious aberrations when it ought to be a matter of curing them" (Van

Gogh 1889d, cf. *Letter to Theo*, n. 801), and where "[...] the majority of the priests seem to me to be in a sad state", awakening ancient ghosts: "Religion has frightened me so much for so many years now" (Van Gogh 1889a, cf. *Letter to Willemien van Gogh*, n. 764).

Vincent van Gogh did not disapprove of biblical inspiration as such, but of the extent to which it reproduces feelings and perceptions that end up distancing one from modern reality rather than helping one to deal with it in a way that awakens appropriate feelings and meditations:

> "it is—no doubt—wise, right, to be moved by the Bible, but modern reality has such a hold over us that even when trying abstractly to reconstruct ancient times in our thoughts—just at that very moment the petty events of our lives tear us away from these meditations and our own adventures throw us forcibly into personal sensations: joy, boredom, suffering, anger or smiling."
>
> (Van Gogh 1889h, cf. *Letter to Emile Bernard*, n. 822)

For these reasons, he refused to paint a "Christ in the Garden of Olives", thinking instead to offer "the correct proportions of the human figure in it", as it might be more useful for people to think about olive trees. Although van Gogh mentions Gethsemane as a metaphor for "terrible illness", he seems to opt for the contribution of literature; in particular, he invokes Ernest Renan (possibly the *Vie du Jésus*) to help him through words to see in the "blue sky and the gentle rustling of the olive trees", among other "explanatory things" that can "turn his history into a resurrection" (Van Gogh 1889a, cf. *Letter to Willemien van Gogh*, n. 764). In this context, the olive trees and the blue sky, and especially the yellow, seem to be a metaphor for his fight against the disease and a way to foster his hope for a cure (Van Gogh 1889b, cf. *Letter to Theo*, n. 776). Then, he painted some pictures with the olive trees motifs, without the suffering of Christ (Figures 6 and 7), but "coloured with solemn tones" (Van Gogh 1890a, *Letter to Theo*, n. 834; 1889g, *Letter to Theo*, n. 820).

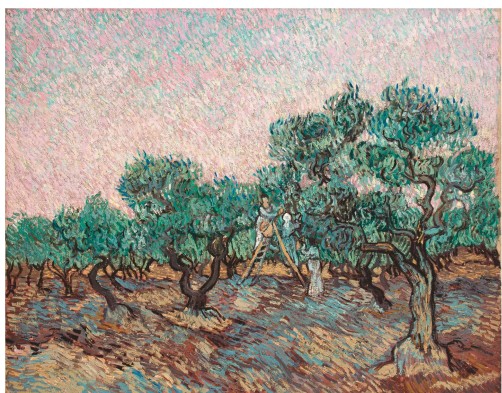

**Figure 6.** Women Picking Olives, 1889.

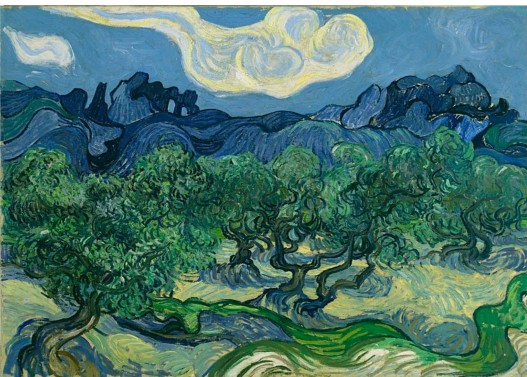

**Figure 7.** Olive Trees in a Mountainous Landscape, 1889.

In the same year, *Pietá* was painted, based in a lithograph (Figure 8) of a painting of Eugene Delacroix (Van Gogh 1889f, cf. *Letter to Theo*, n. 805), however with a new chromatology in which a bluer tone prevails over the light (Figure 9):

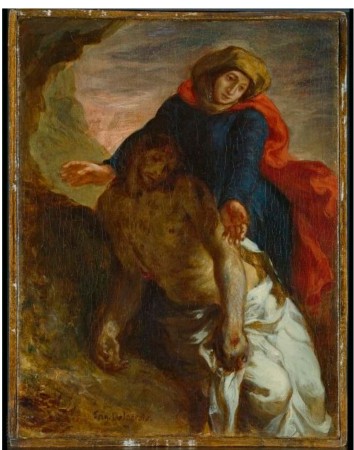

**Figure 8.** *Pietà*, by Delacroix, 1850.

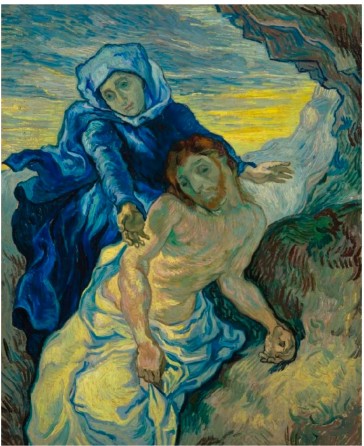

**Figure 9.** *Pietà* (after Delacroix), by Van Gogh, 1889.

In parallel with two other works of van Gogh from the following year, it seems to indicate that this bluer tone may be correlated with the time of trials, of the forces above human life, a time of growth, just as the *wheatfield* (Figures 10 and 11) seems to represent the human condition:

> "[about the wheatfields] Their story is ours, for we who live on bread, are we not ourselves wheat to a considerable extent, at least ought we not to submit to growing, powerless to move, like a plant, relative to what our imagination sometimes desires, and to be reaped when we are ripe, as it is?"
>
> (Van Gogh 1889c, cf. *Letter to Willemien van Gogh*, n. 785)

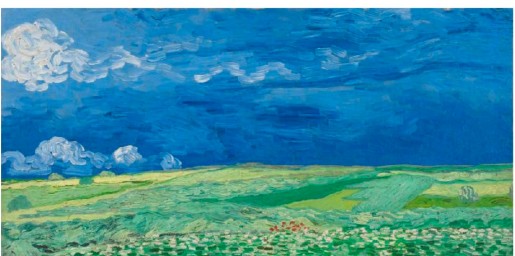

**Figure 10.** Wheatfield under Thunderclouds, 1890.

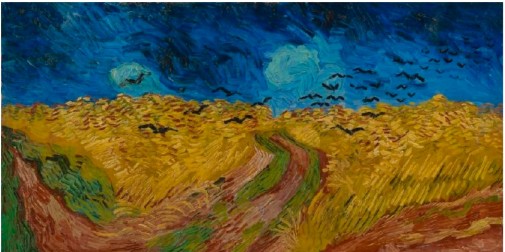

**Figure 11.** Wheatfield with Crows, 1890.

Particularly, the painting *Wheatfield with Crows* (1890) seems to translate the painter's belief that the light is present instead of the darkness (Figure 11). And painting is like a spiritual exercise where light emerges. During two months inside his hospital room, feeling the loneliness, the painter confessed: "It's only in front of the easel while painting that I feel a little of life" (Van Gogh 1889e, cf. *Letter to Willemien van Gogh*, n. 804).

## 6. Theological Chromatology

On 16 May 1890, van Gogh left the asylum after being declared cured by Doctor Théphile Peyron, the director of the institution (Van der Veen 2023, p. 194). In the same month, van Gogh painted *The Raising of Lazarus (after Rembrandt)*, perhaps as a way of symbolising the resurrection of his story, as he had mentioned to his sister during his time of treatment and hospitalisation.

There seemed to be some relationship between the choice of colours and the theological content present in the motifs of his paintings, as if composing a kind of key theological chromatology. A fundamental framework for understanding the theological and chromatological references is the way in which van Gogh re-reads Rembrandt, in his painting, months before his death in 1890. It is important to identify his appreciation for Rembrandt Harmenszoon van Rijn (1606–1669), the great Dutch painter in 17th century. For Rembrandt, religion had a central place in his paintings and in his Calvinist worldview. However, for van Gogh as a Dutch painter from the 19th century, Rembrandt, despite his genius, was only "a painter of portraits" who needs to be "re-created in broad strokes" with "very wide margins" to insert, among other things, "landscapes", "interior scenes" and "philosophical subjects" from the new times (Van Gogh 1888a, cf. *Letter to Émile Bernard*, n. 651).

In this sense, Rembrandt frames the Dutch aesthetic culture in which deeper dimensions must be inserted, dimensions related to the need for a spirituality that meets the modern reality of a new time. In this sense, Rembrandt's work, *The Raising of Lazarus* from 1632 (Figure 12), seems to have been taken as a frame (Van Gogh 1890c, cf. *Letter to Theo*, n. 865) to be expanded by van Gogh's chromatology and revisited in the way of signification of his motif (Figure 13):

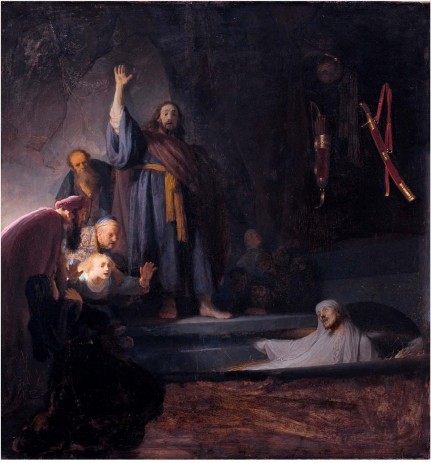

**Figure 12.** The Raising of Lazarus, by Rembrandt, 1632.

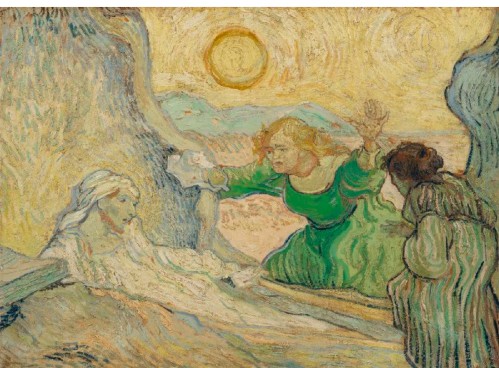

**Figure 13.** The Raising of Lazarus (after Rembrandt), by Van Gogh, 1890.

This substitution of Rembrandt's Christ raises a number of theological issues and points to van Gogh's critique of the hegemonic theological context of his time and the struggle over religious imagery in the endless theological doctrinal clash of that time, between Calvinist Protestant scholasticism and the Catholic scholasticism of the Counter-Reformation, especially with regard to salvation and its social consequences. Rembrandt was not a strict Calvinist, nor did he formally belong to the Dutch Reformed Church, but he was certainly influenced by Calvin's theology and was a careful reader of the *Statenbijbel* (Perlove and Silver 2009, pp. 18–24), the Dutch government-funded translation of the Bible in the 17th century, with annotations by Calvinist theologians, directly from the Hebrew and Greek originals. He also knew the *questio disputata* between the professors at the University of Leiden, where he lived and studied (1615–1619), namely the question known as *De praedestinatione* or double predestination. According to this theory, God, before creating the world, chose to save certain individuals for His own reasons and regardless of any conditions attached to these people, defended by Franciscus Gomarus (1563–1641) and criticised by Jacobus Arminius (1560–1609), in the sense of rethinking the interpretation of orthodox Calvinism, taking into account human freedom in relation to God and the Church. This was the context for the emergence of the Arminians, a group condemned by the synod of Dordrecht (1618–1619) after the death of Arminius in 1609.

Rembrandt's interest as a reader in the work of Jan Philipsz Schabaelje (1592–1656), an important Mennonite writer considered heretical by the official Dutch church, suggests at least some reservations on the part of the Flemish painter about Calvinist dogma. The Mennonite also inspired his work through a series of biblical engravings in a visual representation of the most important stories, teachings and mysteries in all of Scriptures, called *Den Grooten Emblem Sacra* [The Great Sacred Emblems] (Perlove and Silver 2009, pp. 8, 23, 88).

Schabaelje's proposal seems to collaborate with another theological dispute, namely the dispute over the imaginary and how theological images affect spiritual practices, especially with regard to the Counter-Reformation and the importance of biblical images for the contemplative exercises proposed by the Jesuits (Stronks 2012, p. 9), specifically the Spiritual Exercises of St Ignatius of Loyola. Although *Den Grooten Emblem Sacra* was not published until 1646, in 1635 the author published another work using biblical images as a resource for a Protestant version of the spiritual exercises: (*geestelycke oeffeningen*) called *Lusthof des Gemoets inhoudende verscheyden geestelijcke Oeffeningen met noch twee Collatien der wandelende Ziele met Adam em Noach* [The Mind's Garden of Pleasure, Containing Various Spiritual Exercises with Two Dialogues of the Wandering Soul with Adam and Noah] (Schabaelje 1717), which was later published in English as *Wandering soul*.

This influence of correlating the search for an experience of faith and a Calvinist visual exegesis (Perlove 1989, pp. 11–23) of biblical imagery through art results in the dramatic dimension of the salvation of a God who reveals himself obliquely throughout the history of the people of God (Calvin 1813, cf. *Institutes of the Christian Religion II, X, XX*), which is aesthetically expressed in Rembrandt's *chiaroscuro* style, in the struggle of

life between light and darkness, and in this way could be a kind of influence, at least on the dramatic dimension of life. However, the artist's apparent lack of orthodoxy, in line with the iconoclastic tendency of Dutch Calvinism (Freedberg 1986), consolidated since the iconoclastic *Beeldenstorm* movement (1525–1580), meant that Rembrandt was very slow to become part of an official Calvinist aesthetic. The reproduction of his engravings with a precisely religious purpose was only possible with Cornelis Hofstede de Groot's (1863–1930) proposal in 1890 to illustrate the *Statenbijbel* with works by the Dutch painter, following the publication of a French edition of the Bible illustrated in 1866 by Gustave Doré (1832–1883). His proposal resulted in the publication that became known as the *Rembrandt-Bijbel*. Despite being a late edition, the possibility of proposing such an undertaking already indicated a weakening of the iconoclastic tendency in van Gogh's time, in which the 17th-century painter's strong Calvinist belief in the efficacy of faith and the Bible naturally placed Rembrandt in the environment of 19th-century Calvinism.

Consequently, the substitution of Rembrandt's Christ for the sun in van Gogh's Lazarus confers another Christological significance, of someone who distrusts Calvinist institutions, and consequently the way their main theologian interprets the Bible. The dark drama of salvation is replaced in van Gogh by the unequivocal friendship of God that dispels any supposed divine intention of a dramatic play. This van Gogh's significance permeates his entire work and expands the work's own aesthetic significance. Yellow can be seen as the expression of the *Christus Consolator* who accompanied the painter from the time of his fervent piety (Van Gogh 1877a, cf. *Letter to Theo*, n. 104), and who seems to be viewed by the painter as remaining his friendship, just as he did not abandon Lazarus in the darkness of death. Some critics even said that Lazarus in the picture is van Gogh himself.[5]

This paschal logic of life conferred by *Christus Consolator* can be seen as a source of hope and joy. Life, even at in completely nonsensical times, is still capable of meaning, which springs from the moments when the joy of living returns. This bet on meaning despite life's absurd moments is transmitted by this theological chromatology in order to transmit a *je ne sais quoi*, a numinous kind of *I don't know what* of the eternal, an experience of the density of meaning that life carries, which the Dutch painter feels and expresses best with his brush:

> "in a painting I'd like to say something consoling, like a piece of music. I'd like to paint men or women with that *je ne sais quoi* of the eternal, of which the halo used to be the symbol, and which we try to achieve through the radiance itself, through the vibrancy of our colorations" [author's emphasis]
>
> (Van Gogh 1888b, cf. *Letter to Theo*, n. 673)

In this sense, van Gogh's paintings seem to shift the focus from the halo of holiness that was traditionally represented to favouring the holiness of those considered chosen. In fact, Calvin states in his *Institutas* that "Christ, the sun of righteousness, completely illuminated the whole world" (Calvin 1813, cf. *Institutes of the Christian Religion* II, X, XX). But the light that illuminate "minds" (I, V, IV) is related to the acceptance of Christ as Lord and Saviour and is transmitted by faith only (what this means within the doctrinal contours of the Reformed Church) and is limited to God's "elect" (those who have accepted faith) and not to the "reprobate" (II, II, XI). It is here that van Gogh seems to have completely separated himself from his Calvinist heritage, at least in the way that the theology of predestination restricted God's love to his elect. The Dutch painter's vision coincides in this case with the criticism that an important Calvinist theologian of the 20th century makes of what he considers to be a limit in Calvin's theology:

> "It is remarkable how little Calvin has to say about the Divine love. The Divine glory replaces the Divine love. And if he speaks of the Divine love, it is love towards those who are elected. But the universality of the Divine love is denied, and the demonic negation, the split of the world, has in Calvin a kind of eternity, through his doctrine of double predestination. Therefore this is a doctrine which contradicts the doctrine of the Divine love as sustaining everything"
>
> (Paul Tillich 1967, p. 270)

Furthermore, in the 19th century the idea of election was further aggravated by being linked to the notion of prosperity as a sign of divine election, as Max Weber pointed out. Tillich himself recognises how this doctrine of double predestination leads Weber to consider elements of the "spirit of Calvinistic ethics" and how this relates to a "sectarian ethics which is useful for the purposes of investment in the capitalistic economy" (Paul Tillich 1967, p. 270).

It was here that van Gogh went against the grain of double predestination and its capitalist impact on the contradictions of the industrial revolution. His emphasis shifts to the gratuitousness of God's love, which enlightens and sanctifies everyone through the power of the love he awakens in everyone, especially the most invisible, as represented in his paintings in the presence of the Christological sun in the most disfavoured social strata, offering his friendship (Figures 14 and 15):

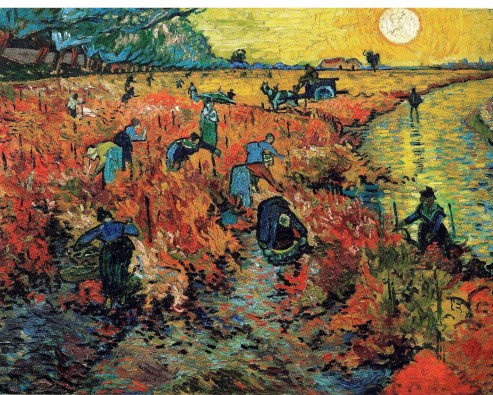

**Figure 14.** Red Vineyard, 1888.

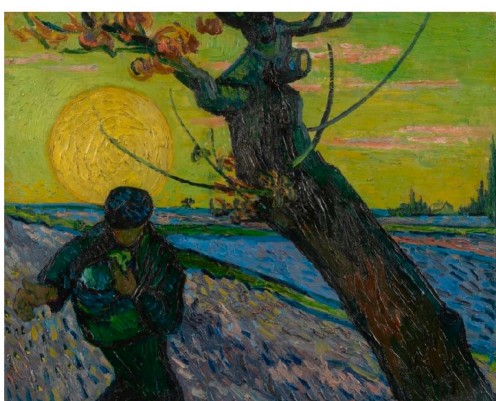

**Figure 15.** Sower at Sunset, 1888.

It is precisely in this context of the link between salvation and prosperity, opened up by the relationship between the capitalist spirit and Calvinist ethics, that it is possible to identify an option for the poor *avant la lettre*, bearing in mind that the use of this concept only appeared in the 20th century in Latin America with the Catholic theologian Gustavo Gutiérrez (1971) and the Presbyterian theologian Rubem Alves (1969), as a critique of neo-liberal policies and structural socio-political relations with a strictly religious mentality. What was the interpretation of a certain sectarian Calvinist spirit in the Protestant world is the equivalent of French Jansenism in the Catholic world, both characterised by a self-referential theology in the guise of a more "spiritual" religion (Groethuysen 1927, pp. 31, 46). The way in which God makes himself known in van Gogh is through a revelation that is revealed in *praxis* (Metz 1999, pp. 246–55), and in its correlation with the *poeísis* (Villas Boas 2016, p. 194), as a hermeneutic of the frontier of a way of thinking, in the face of the new challenges it imposes (Duque 2019).

Such a correlation can be thought of about the application of theological chromatology suggested here in van Gogh's last painting with a religious motif, precisely "The Church at Auvers-sur-Oise" (Figure 16). The picture, whose interior evokes a chromatology of the time of crisis, but inside the building itself, is darker even than the storm that is brewing, in which the institution can be perceived from its self-referentiality, far from the lives of those who suffer most, closed in on its own storms and crises, and still casting shadows, rather than light, on the common ground of life, where light can be found.

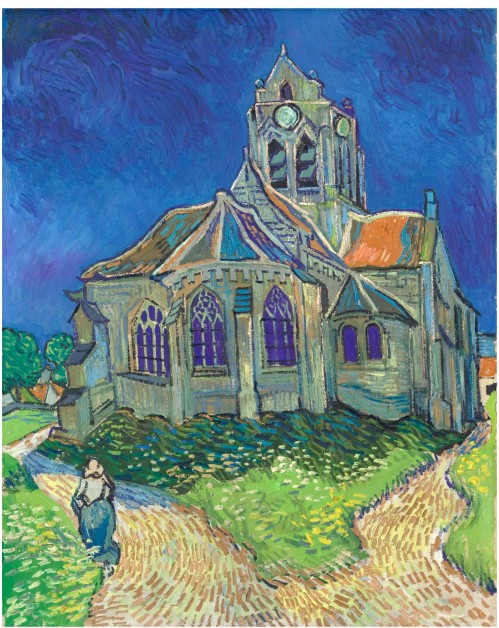

**Figure 16.** Church at Auvers-sur-Oise, 1890.

This soteriological chromatology seems to indicate the friendly presence of Christ especially in those who suffer the most, and of which the perspective of van Gogh in different ways was attentive, and gave visibility to them, inviting a paschal view of resilience despite all injustice, and to those who contemplate the works to realize that everything is connected and supported by the love of God, the source of every form of commitment and ethical resistance. In this way, van Gogh's theological chromatology seems to assume the mission of spirituality to offer meaning for life through the language of colours in a constant existential dynamic of passion, death and resurrection (Figures 3, 9 and 13, triptych shown in Figure 17), or even *enchantment* (awakened by the joy of living), *disenchantment* and *re-enchantment* with life.

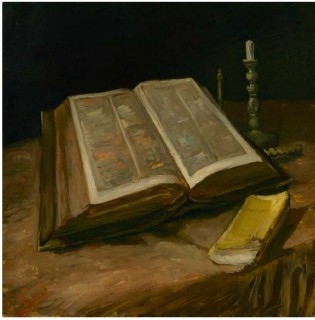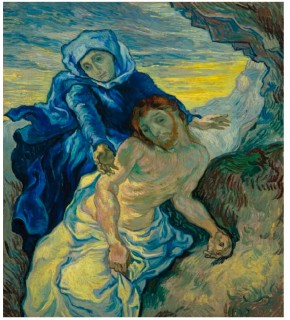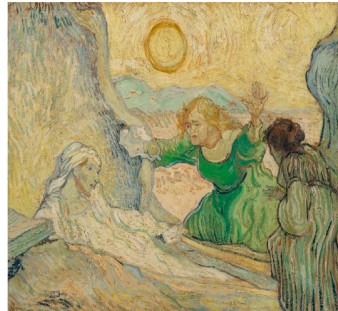

**Figure 17.** Vincent van Gogh's theological chromatology.

This Christological dynamic present in van Gogh's theological chromatology is different from a theodicy where all the absurdities of history come from God's will, but it concerns a theological view about a God that is always present in the life of the people

as a friend, as it evokes the motif of Christ's resurrection of his friend Lazarus, dispelling the darkness with his presence. It is about a God who never abandons his people, just as Christ does not abandon his friends, and it is a source of serenity and maturity to welcome the new possibilities of meaning for life that will come, a way of reconciliation with one's own history:

> "To understand all is to forgive all, and I believe that *if* we knew everything we'd arrive at a certain serenity. Now having this serenity as much as possible, even when one knows—little—nothing—for certain, is perhaps a better remedy against all ills than what's sold in the chemist's."
>
> (Van Gogh 1887, cf. *Letter to Willemien van Gogh*, n. 574)

The Christological reference to the sun at the end of his life (two months before he lost his life) allows one to think of the reactivation of his theological discursive practice, at least in its discursive materiality, which now has a visual materiality. Therefore, for van Gogh if "truly life is a fight" and "by fighting the difficulties in which one finds oneself, an inner strength develops from within our heart, which improves in life's fight (one matures in the storm)", then "God's finest gift" is that "He be not far from every one of us" and "that our light comes in the darkness of life and of the world". Hence, "if we feel an eye watching us, as it were, then it is good to gaze upward sometimes as though seeing Him who is invisible" (Van Gogh 1877e, cf. *Letter to Theo*, n. 133). Ultimately, the struggles for just causes are opportunities to discover God's presence in our personal and collective history. Moreover, it is necessary to desire to look and remain looking at love in order to let it germinate the fruits of life, and in this way light can be found despite the darkness:

> "Now comparing people with grains of wheat—in every person who's healthy and natural there's *the power to germinate* as in a grain of wheat. And so natural life is *germinating*. What the power to germinate is in wheat, so love is in us."
>
> (Van Gogh 1887, cf. *Letter to Willemien van Gogh*, n. 574)

Thus, the dynamics of fighting for a life that is able to find beauty and meaning despite its moments of suffering and darkness are painted in Vincent van Gogh's kairological yellow "symbolizing gratitude in the rustic sunflower" (Figure 18), which lives an existence facing the sun, facing the love and peace, rather than emphasising the fragility of someone who suffered severe mental disorders with precarious treatment at the time; one can see much more of a struggle up to the last days of life, engaged in the task of humanising the human being (Van Gogh 1890b, cf. *Letter to Willemien van Gogh*, n. 856):

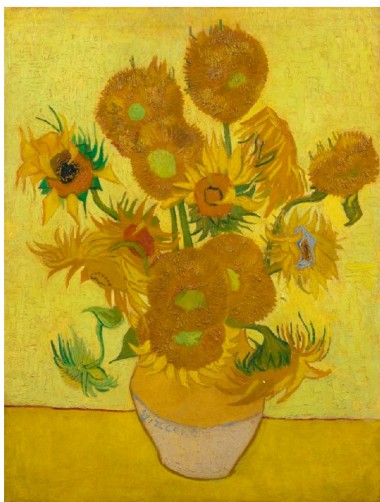

**Figure 18.** Sunflowers, 1889.

It can also be said that, evoking the motif of the sun and the image of the sun seen from the angle of the star-sized coal miners, no matter how dark the darkness, social or

interior, the light of this friendship does not stop shining, without demanding conditions, but it simply shines like the Sun. The painter's work is a kind of invitation to insist on looking at what apparently does not offer beauty to our eyes, to let the seed of faith grow as faith for life too. And this paradox of light over the night, of meaning over the absurdity of living, can be seen in his painting *Starry night* (Figure 19). Said the painter: "*When all sounds cease—God's voice is heard—Under the stars*" (Van Gogh 1877c, cf. *Letter to Theo*, n. 119), and therefore, darkness is not the place of resignation, but of discernment, resilience and hope.

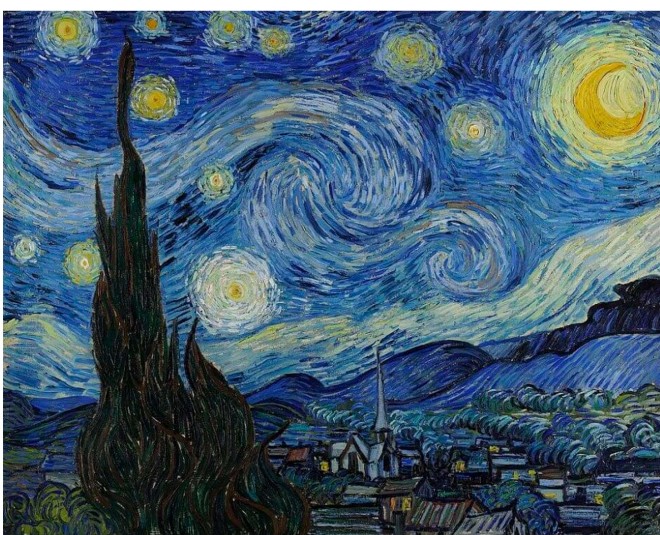

**Figure 19.** Starry night, 1889.

### 7. Conclusions

Vincent van Gogh's statements on life can be seen in the materiality of his letters transformed into works of art. Painting for van Gogh could be seen as a spiritual exercise to see the presence of light instead of times of darkness, and an exercise to see the light through the lens of God's love for mankind. Sometimes the light shines like the sunlight in summer, and sometimes it is only possible to see the daylight about the size of a star shining in the dark night, but the night does not change the sun, only the view of the beholder. The stars in van Gogh's Starry Night can be seen as a sacrament of light, remembering and announcing its presence.

The density of van Gogh's theological chromatology is very rich, thinking of literature and art in dialogue with spirituality, combining existential questions and commitment to those who suffer most.

One of the most important elements, however, is the way in which a religious subjectivity is produced in the genius of contemporary painting. This implies an agonising process of overcoming the incompatibility between the social sensibility that unfolds from a religious spirituality and a new subjectivity but does not coincide with the normative theology of the same ecclesial context in which the sources of the religious motivation of the protagonist of this experience are found.

It is still possible to identify some traces of his experience in the Borinage, of the dark side of the Industrial Revolution in the young aspiring pastor, such as the emergence of human suffering in wretched conditions, along the distance of the ecclesial community from this misery. The great disappointment with the Church unfolded the search for God in a search for self-knowledge in a shared condition with the poorest people, the miners, represented by the tension between the presence of light, symbolising life, and the darkness, symbolising the exploitation to which they were subjected, in an awakening of a vocation that recalculated the route from the theologian to the painter.

Whether one rejects or canonises van Gogh in relation to Christianity, his life and work can certainly be seen as a heterological expression of theology, a Trojan horse in religious

culture that, by directly influencing culture, also reveals something of the emerging religious sensibility of modernity.

Despite his trauma with the Church, there remains in him a deep sense of spirituality, symptomatic of the times to come, which does not exactly imply a crisis of faith, but a crisis of faith in the institution. It also points to a growing modern need in which faith tends to be experienced in the dynamic of spirituality, in the search for meaning and practical wisdom for everyday life, rather than in the defence of a set of doctrines or normative schemes. This certainly implies a rethinking of moral requirements as a condition for welcoming people, a rethinking of community as that which accompanies people and their suffering, like Christ in the image of Lazarus, who is always a friend.

The dynamism of the contemplative exercise of seeing beauty where it is not obvious can be seen as the result of this spirituality, as a view of life from the perspective of a God who loves the whole of creation. But it requires a literary mediation that spares no effort to present life in its ambiguity, far from a dialectical logic that transforms complexity into reductive polarities and encloses the very literary dimension of the Bible and its poetic beauty in normative idealisms.

Moreover, van Gogh's political spirituality related to people not from dogmatic constraints or theoretical abstractions, but in a commitment to those who suffer the most, thinking of history in the wounds of the people, giving visibility to the forgotten of Borinage, at a time when no social media cared about them. In van Gogh's attempt to give them visibility, he also ended up immortalising an attitude that still resonates as a challenge to a theology that thinks in commitment to those who suffer most.

What is most impressive is that this brilliant painter not only developed his spirituality and theological reflection without religious support, but also faced all the discredit that could have been associated with a space of unreason, social exclusion and religious impertinence. In this sense, his process of existential overcoming, the "wall to be crossed", was not only about his personal challenge, but coincided with the process of overcoming the mentality and a discursive practice of an epoch, in particular the dialectical logic of disqualifying others for their alleged form of unreason, due to ideological non-alignment. Moreover, in van Gogh's case it was not just a question of ideological non-alignment, but a double disqualification due to his undiagnosed illness and subsequent suicide. Still on this subject, a highly symbolic gesture of how ahead of its time the van Gogh phenomenon was can be seen in the fact that the refusal to associate his name with the religious community in the 19th century was completely overcome by the fact that one of the two works called Pietà was installed in the Vatican Museum in the 20th century.[6]

Empathy with the suffering of the weakest, the desire to love more than to be loved and a reading of the Bible in the dynamic of parables through the lens of modern literature are together a recipe for finding the joy of living. This perspective dwells in the bet that life is capable of meaning, despite all the absurdity and our own contradiction, and this is the condition for insisting on seeing the light, despite the moments of darkness in which one lives, and which van Gogh's painting tries to reveal. In the colours, the painter seems to seek the presence of God in life, who, like the sun, never ceases to be present and to dynamize all the vitality of the world, even when it is not possible to see him.

In this way, van Gogh's perspective can be assigned to an ethical turn. In this way, ethics represents as a social practice what poetics and art represent as discursive and aesthetic practices, namely the opening of a space that does not need to be authorised by the established order of reality, a seed of possibility in the cracks of the walls of impossibility that creates bridges for hope. The seemingly impossible unfolds not only as a contestation, but potentially and fruitfully in culture as the emergence of the impossible that inhabits desire, the engine of an expansion of collective intelligence through empathy with the pains of an era, with hope in the cracks not yet opened in history but already inhabited by desire. In this sense, a theological critique could emerge tactically, through a new sensitivity to the suffering of an era, before theological claims.

Finally, the American poet Abigail Carroll has dedicated a poem, Dear Lover of Light, to Vincent van Gogh for his ability to look for the beauty in life even in such difficult times and places. The poem is part of a book project called Letters to Saint Francis from a Modern-Day Pilgrim, which not only sees van Gogh as a modern-day pilgrim, but also introduces him to Saint Francis as one of his modern-day disciples:

> "There lived a priest
> so in love with light
> it drove him mad.
> Paint was his thing.
> When he could no longer
> preach, he hopped a train
> south, took up a brush,
> turned zinc and lead
> and chrome
> into gaudy, wild-
> petaled ambassadors
> of the dawn. He slapped
> stars as big as brooches
> on the sky, danced
> crows across bowing fields
> of wheat, exalted a bowl
> of onions, a bridge, a pipe,
> a chair, a bed. Postmen
> and prostitutes
> were his friends—
> so too were irises,
> almond trees,
> windmills,
> clouds. Francis,
> if you think of a painting
> as a kind of song, he too
> canticled the sun." (Carroll 2017)

**Funding:** This research received external funding from FCT—Fundação para a Ciência e a Tecnologia, Portugal Government through the Scientific Employment Stimulus Programme nº FCTEECC00062019 which this article was developed in the context of the research carried out in the working group "Theopoetics as public theology: an archaeology of the discursive and social practices of theology" in the project "Common home and new ways of living interculturally: Public theology and ecology of culture in pandemic times. The project is co-financed by CITER—Research Centre for Theology and Religious Studies of the Universidade Católica Portuguesa (UCP) and the Global Educational Pact Bureau of the Pontifícia Universidade Católica do Paraná (PUCPR).

**Data Availability Statement:** No new data were created or analyzed in this study. Data sharing is not applicable to this article.

**Acknowledgments:** We are grateful for MDPI editors for the invitation to publish this article as part of the "Religions 2022 Young Investigator Award" and for FCT to afford the research project which this article came from. I would also like to thank my colleagues at the Faculty of Theology at the University of Kiel, Germany, for the rich dialogue where this article was originally presented as a lecture and later expanded and developed into this final article.

**Conflicts of Interest:** The author declares no conflicts of interest.

## Notes

1    The letters by Vincent van Gogh used in this article are mainly taken from the collection "Vincent van Gogh: The Letters", edited by Leo Jansen, Hans Luijten and Nienke Bakker and made available by the Van Gogh Museum (Jansen et al. 2009a). Available on line: https://vangoghletters.org/vg/letters.html (accessed on 1 June 2022). In some cases, use will be made of certain collections,

either because of the editors' comments or because the material contains information considered valuable for this article. This is the case, for example, with the anthology of letters by and about the painter edited by the British-American poet Wystan Hugh Auden, called "Van Gogh: A Self-Portrait", from 1961 (Auden 1961).

2   Van Gogh's collection of letters has been published in parts, sometimes arranged according to the recipient of the letter. Even the complete collections have been partially published, depending on access to the material, including the issue of differences between originals and copies, requiring concordance work to agree on the publication of the letters. In this sense, Heinich's (1996) work uses some collections that do not correspond to the current classification of the Van Gogh Museum for the publication of the complete letters. For example, when the sociologist quotes the painter's letter to his brother in which he makes the above-mentioned statement about religion, the museum's collection classifies it as Letter No. 691, not 543. The same happens with other quotations. The author's comparison of quotations with the Van Gogh Museum's collection must therefore be based on the addressee and the date. For the purposes of this work, we will keep Heinich's quotation as it appears in her work.

3   Orig.: ye

4   Orig.: dull.

5   The suggestion is endorsed by the Van Gogh Museum itself: "He probably identified with Lazarus in the tomb. That would explain why he gave the figure a red beard." Cf. Collections, The Raising of Lazarus (after Rembrandt): https://www.vangoghmuseum.nl/en/collection/s0169V1962, accessed on 20 May 2022.

6   Vincent van Gogh, *Pietà*. Musei Vaticani. Collection of Modern and Contemporary Art. Donated by the diocese of New York, 1973. https://www.museivaticani.va/content/museivaticani/en/collezioni/musei/collezione-d_arte-contemporanea/sala-2--van-gogh--gauguin--medardo-rosso/vincent-van-gogh--pieta.html, accessed on 27 December 2023.

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
