# Peer review of "Vincent van Gogh’s Theological Chromatology: A Critical Reader of the Bible from His Option for the Poor Avant la Lettre"

_religions, doi:10.3390/rel15040425_

Round 1

Reviewer 1 Report

Comments and Suggestions for Authors

This is an excellent topic and approach to theology, the visual arts and religion in everday life. Van Gogh's missionary zeal, so to speak, is maintained as the perception of hope and the holy even in the midst of darkness and suffering and beyond the language and scope of institutionalized religion and theology. The dialogue between visual arts and literature (Zola, the Bible) is also very interesting, as is the extensive use of van Gogh's letters as a source. I wonder whether they subtitle could not rather be something like "a theological chromatology sensitive to the poor" than a reference to the as such in the text little discussed and a bit worn-out (although still relevant) "option for the poor". 

A more thorough contextualization of religion in Dutch Protestantism of Van Gogh's time would be helpful both to understand his own point of departure and the surroundings that did not welcome him later. This would also greatly help to better understand this somewhat engimatic sentence in the conclusion: "This implies an agonising process of overcoming the incompatibility between the social sensibility [sic, sensitivity] that unfolds from a religious spirituality but does not coincide with the normative theology of the same ecclesial context in which the sources of the religious motivation of the protagonist of this experience are found." (p. 20, lines 782-785)

The distinct theological chromatology of van Gogh, not least in contrast to Rembrandt, could be more deeply conceptualized. While both pictures (duly authorized for [re-]publication in the journal, I presume) and text do indicate the importance of lightplay and namely of yellow, there seems to have to be a bit more to a "chromatology" in terms of the use of colours in a theological perspective.

p. 1 line 25 – sold for more than 100 million Euros -> reference?

p. 1 line 27 – suggestion: “who died at 37 years of age”

p. 1 line 27 – suggestion: “suspected of having taken his own life”, and “within a century”

p. 1 line 32ff. – give references

p. 3 line 130 give references

p. 4 lines 178-80 are from the template and should be erased

p. 6, lines 265-66, you could mention this is a paraphrasis of the biblical book of Ruth 1,16

p. 6, line 271: is something missing here? Or should the sentence end on “the poorest” with a full stop? And what follows in lines 273-276 repeats what has just been said, but expands it. Verify and correct.

p. 8: How did the family react to van Gogh’s disapproval by the Consistory? After all, this was a lineage of pastors?

p. 9, lines 444ff. – in my understanding, the quotation does not exactly corroborate what the lines before that introduce it state – certainly it is not about human goodness, but about the belief that whatever is good comes from God, and what is not does not come from God (but from or at least through humans…). Neither does it speak about transformation.

Comments on the Quality of English Language

The text is well written, but apparently in another language than English in its original and then translated, possibly using Google translator, which is generally reasonable but made some parts rather odd to read or even difficult to understand. A thorough revision is needed. I am giving a good number of correction suggestions below but insist this list is not exhaustive.

Harmonize as “Van Gogh” or “van Gogh” (e.g. p. 6 lines 251 and 252) - I suggest "van Gogh". Name the brother and also the sister when they first appear in the text.

p. 5, line 244: “exchanged by Van Gogh” – either indicate between him and whom or substitute for “written”

p. 5, note 2: Nathalie Heinich, and thus “her work”

p. 6 lines 249-50: revise the phrase

p. 6 line 257: suggestion:  “van Gogh was 24 years old”

p. 6, lines 282-283: “Thomas a Kempis”, not “de Kempis”, in a Latinized manner, or “Thomas of Kempen” in an Anglicized manner

p. 6, line 292: specify the “he” – on the whole, the sentence is not clear, not in itself an not in its connection to the earlier paragraph

p. 7, line 323: it should read “van Gogh’s dedication”

p. 7, line 350: it should read “one can find”

p. 8, line 371: it should read “the great recognition he had from”

p. 8, line 373-374: rephrase: “due to an alleged lack of a ‘talent for speaking’, even”

p. 8, line 391: “stoning” should be, at least, in quotation marks, if it be mentioned at all, as van Gogh was certainly not literally stonedp. 8, line 400: maybe, rather than “was feed” (which anyway should be “was fed”), “was nourished”

p. 9, line 432: it should read “ a good friend of van Gogh’s”p. 11, line 494: it should read “On the other hand”

p. 11, line 498: “contiguity” – do you mean “contingency”?

p. 11, line 508: it should read “and his depressive pessimism”

p. 12, line 518: “abbé” here means a rector/parish priest, not an abbot; one could also say “Father Horteur”

p. 12, line 536: it should probably read “in which he talks about Zola, and makes a comment on the Bible” (or simply “and comments”)

p. 12, line 537: rephrase “those who just sit [in?] melancholy” and then “he [Jesus] is not here, he is risen” [quoting the angel in Mt 28,6]

p. 13, line 560: it should read “Candide”

p. 13, line 581: drop “to”, to read “told his sister years later”

p. 15, lines 630-32: the subject of this sentence is not clear, please revise. Also, it should read ”tone”

p. 16, line 646: “belief” rather than “believe”

p. 16, line 665: it should read “Dutch painter from the 19th century”

p. 17, line 685: it should read “death”, rather than “Death”

Author Response

Thank you very much for reading the article carefully and for your valuable comments, which have undoubtedly helped a considerable improvement.

Attached are the comments on the suggestions for improvement, all of which are very welcome.

Warm regards, 

The Author

Reviewer 2 Report

Comments and Suggestions for Authors

The article is part of the attempt, already pointed out by Michel de Certeau, to carry out an archaeology of religion in the space of 19th century knowledge. As part of this endeavour, the author seeks to establish a relationship between theology and the artistic-pictorial expression of spirituality based on some paintings with religious motifs by Vincent Van Gogh. On the one hand, he emphasises the self-referential normative theology (p.1, line 41) of the 19th century and, on the other, the spirituality glimpsed in the wisdom of contemplative experience and in the theological concern of some of the Dutch painter's paintings. The author considers that, in addition to philosophy and the human sciences, which are the objects of Michel Foucault's archaeology of knowledge, religion and its different expressions (theology, spirituality, iconography, hagiographic literature) are also a constituent part of the structure of Western culture (p. 1, line 41).

The author invests in the idea of theological chromatology in Van Gogh's painting, contrasting the dark colours, signs of human suffering, and the light colours that symbolise the will to live. The ambiguity of this play of colours is a sign of the ambiguity of the human condition itself and of divine goodness in creation. This chromatology is presented as a heterology, in the sense of something that breaks with the normative idealism of an era. The author favours the Dutch painter's critical inspiration in the "preferential option for the poor" (avant la lettre) identified among the coal miners of Borinage, Belgium. The aim of the article is "to identify, through the intersection of his letters and paintings, which discursive practice critically breaks with a particular model of theology, and to help think about how such a perspective and tension is relevant to understanding the present." (p. 3, lines 146-147).

The originality of the study consists in postulating that theological knowledge can be the object of an archaeology, that is, the identification of discursive practices whose horizon of understanding is delimited in certain spaces of knowledge, and in which artistic expression can be thought of as a heterology, that which tends towards the limits and possibilities of what is said and what is seen, of what is enunciable and what is visible in an era. In the conclusion of the article, the effectiveness of this Foucauldian strategy could have been taken up again, in order to assess whether, from an archaeological point of view, the visible and enunciable works that portray the "option for the poor" really break with a self-centred theological knowledge, thus announcing the attempt at another disposition of this knowledge in the second half of the 20th century, which is that of liberation theology. In this sense, the discursive practice observable in Van Gogh's painting could be thought of as a "threshold" that would anticipate the archaeological soil from which liberation theology would emerge.

One of the central elements of the archaeological method is that it does not dispose of the knowledge of an epochal "mentality", based on the individual and conscious will of its authors. This is why Foucault, for example, places thinkers such as Bopp, Cuvier and Ricardo in the domain of the modern episteme in the same framework in which they can be understood. In archaeology, authors are evoked by the "author-function", i.e. this method emphasises that what they think is part of the same archaeological network of an era. However, characters from literature and the arts often appear, tensioning the rigidity of these archaeological networks, such as Don Quixote, Sade and Manet. In the field of theology, the author very originally locates Vincent Van Gogh based on his "author-function", whose discursive practices do not belong entirely to the soil of the 19th and early 20th centuries, just as they announce the religious and theological motifs that are part of the second half of the 20th century. As well as being a kind of "Trojan horse" in 19th century culture, Van Gogh's painting also reveals something of the religious sensibility of Modernity. (p. 21, line 796-7).

Below, we point out some aspects that may make it easier for the reader to understand some of the concepts used in the article. On p. 4, line 171, the author writes: "it is possible to identify the emergence of a political spirituality in Van Gogh's religious experience". It would be interesting to explain more about where the notion of "political spirituality" comes from, what it refers to and whether, in the case of the religious motifs in Van Gogh's paintings, it really is a political spirituality. As it is the only occurrence in the text, it would be worth considering its use, given that it has a unique meaning in Foucault's thinking.

A similar observation can be made about the notion of "critical attitude". It appears in the abstract and only once in the article (p. 4, line 169). It would be interesting to emphasise that Van Gogh's life and work can be examples of the art of not being governed in a certain way in the face of the ecclesial - institutional - insistence on the government of behaviour that belongs to pastoral power as re-signified in Modernity. Therefore, his life and work exemplify this notion, since the criticism is not made "from the outside", but within the framework of pastoral government itself. Furthermore, it is important to know whether the critical attitude is identified with the "criticism" (p. 2, line 64) referred to by Michel de Certeau. Not all "criticism" has the meaning of "critical attitude", since, unlike the critical attitude, it is often carried out as a normative perspective from outside what is being criticised.

Finally, the author states on p.2, lines 59-60, "De Certeau's tactics operate differently from Foucault's strategies, moving from frontal confrontation to installing itself as a 'theology of difference' within the dominant discursive regularity and subverting it from its everyday life, silently producing a cultural revolution". It would be necessary to ponder this statement. As Foucault's "critical attitude" is a "plural art", it means that it doesn't only come about through a "frontal confrontation". Instead, it gives rise to various aspects of "everyday life", including the strategy of the attitude of obedience as "bad obedience", rather than a frontal confrontation which would be to refuse to obey. It's worth pondering to what extent the theoretical stance on creative resistance that we see in the mystic, the poet and the artist are thought of in different ways in Foucault and Certeau, or if, on the contrary, they are identified, albeit through different languages.

In general, this is a very original article in terms of analysing its subject, with a well-defined theoretical framework that, in my opinion, expands the theoretical and practical pretensions of an archaeology of religion postulated by Certeau, as well as adequately realizing the Foucauldian relationship between the author-function and discursive practices, indicating the extent to which the visible (painting) can be tensioned in the face of the enunciable (the order of discourse of normative theology) through Vincent Van Gogh's way of life and paintings with religious motifs.

For these reasons, I recommend the article for publication.

Author Response

(The authors gave the same response as above.)

Reviewer 3 Report

Comments and Suggestions for Authors

To me, this article, is not yet in a sufficiently final form to be considered publishable. The author has clearly done an enormous amount of research and considered the subject in tremendous detail (this is commendable). Nevertheless, what I see here is not very well synthesised and organised. The writing itself needs quite a bit of attention, too. The basic structure of the paper is alright, potentially, but the content needs to be much more carefully thought through and rounded off.

My recommendation, therefore, is that the article should be rewritten and polished before being reconsidered for publication.

My main suggestions would be the following:

First, the article is framed very early on via the work of Foucault. This is discussed in some detail but the meaning and significance of what is said in the earlier part of the article is unclear given everything that follows. I have serious problems with Foucault’s univocal understanding of power, recently critiqued by Byung-Chul Han, and so I would say that a much more critical and nuanced view of his interpretation of madness in relation to power would help. However, that said, my recommendation in the end would be to ditch Foucault completely. His politics and ideological commitments seem to me to be totally out of line with van Gogh’s existence and, more importantly, with his theology. I don't think Foucault offers sufficient illumination for the concerns presented in this article. Looking at Vincent’s life from the outside (via Foucault) seems wrong given how the rest of the paper deals with his existential theology.

Second, considering that the many, many block-quotations of text suggest to me a lack of synthesising and processing, I would suggest that the author takes a careful look at what she or he wishes to argue in order to make a much more careful selection of key ideas from the quoted texts. A closer reading of the quoted texts in relation to Vincent’s life and art is needed. One reason for my suggesting this is that in a few places, the “interpretation” (and some of the deductions, etc) of the quoted texts doesn’t align well with the texts themselves. One prominent example is between lines 405-428. The text quoted reveals Vincent’s idiosyncratic speech pattern (something detectable throughout his writings and which many who knew him commented on — he spoke rapidly with a sort of circular ‘logic’). Then there is the ‘interpretation’ starting in line 420. But I can’t actually see how the author is getting such an interpretation from the quoted text. The risk is too much eisegesis and not enough exegesis.

Comments on the Quality of English Language

The second to opening sentence (lines 25-30) is one warning of some of the writing problems to come. Apart from being excessively wordy here, several concepts are blurred by being so rapidly juxtaposed. It seems, for instance, that mental illness and Vincent's failure as an artist are conflated and, well, this is misleading. There is some sloppy repetition between lines 267 and 277, and often historical events are written about in the present tense. And this is just to name a few direct instances. In many places the article is easy to read and flows well but this is not consistent throughout.

Author Response

(The authors gave the same response as above.)

Round 2

Reviewer 3 Report

Comments and Suggestions for Authors

I am grateful for the author's careful consideration of my comments. The article has been nicely improved and I would recommend that it be published.